# Training Dynamics Explain Safe Early Exits in Diffusion Language Models

## Abstract

Supervised fine-tuning of diffusion language models induces structured neural representations that persist after training and can guide inference. We show that optimization dynamics leave behind actionable signals: aggregating AdamW moment trajectories on Low-Rank Adaptation (LoRA) parameters yields a *Reasoning Representation Map* (RRM), and monitoring its alignment with token activations defines a *Representational Alignment Distribution* (RAD). Our central contribution is to explain not merely that early termination is possible, but *why it is safe*. We prove that small matched-support Kullback–Leibler divergence across consecutive denoising steps bounds multi-step total variation drift, yielding certificates for stability and a no-flip guarantee for predicted tokens. Under mild contraction assumptions, these local guarantees extend globally, showing that RAD stability directly governs inference stability. Building on this theory, we present the **Training-Refined Adaptive Computation Exit (TRACE)** algorithm, which halts generation once RAD stability persists. TRACE reuses lightweight optimizer metadata, requires no retraining or architectural changes, and consistently reduces denoising steps while preserving accuracy. More importantly, it provides the first principled account of why training dynamics encode certifiable inference signals. Our results demonstrate that optimizer states, often discarded after training, encode interpretable structure that links training and inference, enabling adaptive computation with provable safety.

## 1 Introduction

The link between training dynamics and learned representations remains poorly understood, despite its importance for interpretable and controllable AI. During supervised fine-tuning, optimization procedures adjust parameters in task-specific ways, creating structured patterns in parameter space. Yet current deployment pipelines discard this information after training, treating the final weights as a black box.

We show that supervised fine-tuning of diffusion language models induces interpretable neural representations that persist beyond training and can guide inference. Specifically, the AdamW optimizer's moment estimates identify parameters that consistently receive strong, aligned updates (Loshchilov & Hutter, 2019; Kingma & Ba, 2015). Aggregating these patterns yields a *Reasoning Representation Map* (RRM) that encodes the model's computational structure.

These representations are not only interpretable but actionable. Monitoring them during inference produces *Representational Alignment Distributions* (RADs) whose stability certifies when computation has effectively converged. This enables adaptive early exits that reduce denoising steps by 11–68% without loss of accuracy. Our central question, however, is not whether such exits are possible, but *why they are safe*. We show that small matched-support KL divergences across consecutive steps bound multi-step total variation drift, yielding certificates for stability and a no-flip guarantee for predicted tokens. Under mild contraction assumptions, these guarantees extend globally, demonstrating that RAD stability directly controls inference stability.

Building on this insight, we introduce the **Training-Refined Adaptive Computation Exit (TRACE)** algorithm, which halts generation once RAD stability persists for $\Omega$ steps under a KL threshold $\delta$. Unlike prior adaptive-computation methods based on heuristics such as confidence or calibration, TRACE leverages training-time signals that are lightweight to store and intrinsically tied to how capabilities were acquired.

Our framework is general, but we focus on mathematical reasoning tasks (MATH500, GSM8K, Countdown, Sudoku), where intermediate structure allows clear evaluation. We quantify representational behavior using alignment concentration, the late-step change ratio, and certified-stop rates.

**Contributions.** (1) We formalize RRMs from optimizer trajectories and define RADs for inference. (2) We prove matched-support KL→TV certificates (with contraction) that guarantee argmax invariance and Lipschitz stability, providing the first principled explanation of why RAD stability enables safe early termination. (3) We validate this framework on four reasoning benchmarks, showing that certification behavior aligns with RAD dynamics and explains observed efficiency gains.

**Paper outline.** The remainder of this paper is organized as follows. Section 2 introduces Reasoning Representation Maps and defines the Representational Alignment Distribution (RAD) used at inference. Section 3 develops our theoretical framework, proving stability certificates based on matched-support KL→TV bounds and contraction assumptions. Section 4 presents empirical evaluations across four reasoning benchmarks, analyzing diagnostic correlations and certified coverage. Section 5 situates our work in the broader literature on training dynamics, adaptive computation, diffusion models, and efficient inference. Section 6 discusses implications for practice and mathematical reasoning, and Section 7 concludes.

## 2 Reasoning Representation Maps from Training Dynamics

### 2.1 Representation Readout from Training Dynamics

Consider a pre-trained diffusion language model augmented with Low-Rank Adaptation (LoRA) modules in the Query, Key, and Value projections of each Transformer block (Hu et al., 2022). During supervised fine-tuning on reasoning tasks, only these LoRA parameters receive gradient updates. Each LoRA module consists of matrices $A \in \mathbb{R}^{r \times d_{\text{in}}}$ and $B \in \mathbb{R}^{d \times r}$ implementing a low-rank update to the corresponding projection weight matrix, where $d$ is the output dimension and $r$ is the rank.

For a LoRA-B matrix $B \in \mathbb{R}^{d \times r}$, the AdamW optimizer maintains exponentially weighted moving averages of gradients and their second moments (Loshchilov & Hutter, 2019). At training step $k$, these evolve as

$$M_{k,B} = \beta_1 M_{k-1,B} + (1-\beta_1)G_{k,B}, \qquad V_{k,B} = \beta_2 V_{k-1,B} + (1-\beta_2)G_{k,B}^{\odot 2}, \qquad (1)$$

where $G_{k,B} = \nabla_B \mathcal{L}_k$ is the gradient tensor of the training loss at step $k$, $\beta_1, \beta_2 \in [0,1)$ are decay rates, and $\odot$ denotes element-wise operations. The element-wise update magnitude is then

$$U_{k,B} = \frac{M_{k,B}}{\sqrt{V_{k,B}} + \varepsilon}, \qquad (2)$$

where $\varepsilon$ ensures numerical stability. Here $M_{k,B}$ encodes the prevailing direction of recent updates, while $\sqrt{V_{k,B}}$ reflects their stability over time.

**Implementation note.** In standard Adam/AdamW, bias-corrected moments $\hat{M}_{k,B} = M_{k,B}/(1-\beta_1^k)$ and $\hat{V}_{k,B} = V_{k,B}/(1-\beta_2^k)$ are used in the parameter update. For the representation readout we found both the corrected and uncorrected forms to yield nearly identical $u$ after aggregation (Appendix D); unless otherwise stated, we report results with the uncorrected averages in Eqs. equation 1–equation 2.

To create a stable representation of parameter importance patterns, we aggregate the update magnitudes across all $K$ fine-tuning steps:

$$\bar{U}_B = \frac{1}{K} \sum_{k=1}^{K} U_{k,B} \in \mathbb{R}^{d \times r} \tag{3}$$

This tensor captures which elements of the LoRA-B matrix consistently received strong, directional updates during training. Parameters with large values in $\bar{U}_B$ encode core reasoning patterns, while those with small or oscillating values contribute less to the learned capability.

To enable comparison with token-level activations, we reduce $\bar{U}_B$ to a feature-aligned vector through row-wise energy:

$$u[p] = \left\| \bar{U}_B[p,:] \right\|_2 = \sqrt{\sum_{j=1}^{r} \bar{U}_B[p,j]^2}, \quad p = 1, \ldots, d \tag{4}$$

This reduction preserves the total update magnitude each output dimension received across the low-rank components, creating a $d$-dimensional signature we term the Reasoning Representation Map. We subsequently $\ell_2$-normalize $u$ so that the RAD in Eq. equation 6 is invariant to any global rescaling of $\bar{U}_B$; for subspace variants $U$ we orthonormalize columns.

## 2.2 Representational Alignment Distribution

At inference time, we assess how token activations align with the learned representation map. For each visible token $s$ at denoising step $t$, we compute the cosine similarity between its activation $f_s^{(t)} \in \mathbb{R}^d$ and the representation map $u$:

$$\text{Sim}_s^{(t)} = \frac{\langle f_s^{(t)}, u \rangle}{\| f_s^{(t)} \|_2 \| u \|_2}. \tag{5}$$

These alignment scores are converted to a probability distribution using a softmax with fixed temperature $\tau_{\text{blk}}$:

$$P^{(t)}(s) = \frac{\exp\left( \text{Sim}_s^{(t)} / \tau_{\text{blk}} \right)}{\sum_{i \in S_t} \exp\left( \text{Sim}_i^{(t)} / \tau_{\text{blk}} \right)}, \quad s \in S_t, \tag{6}$$

where $S_t$ denotes the set of visible tokens at step $t$. This *Representational Alignment Distribution* (RAD) is the central quantity monitored by TRACE: Algorithm 1 halts once RAD stability is observed across consecutive steps.

## 3 Theoretical Characterization of Representation Stability

The Representational Alignment Distribution (RAD) defined in Section 2.2 is the central quantity used by TRACE to decide when to exit early. In this section we establish theoretical guarantees showing that stability of RADs across consecutive steps implies stability of the model's predictions, thereby providing a principled foundation for the stopping rule in Algorithm 1.

**Conventions.** All logarithms are natural. For any distributions $p, q$ on a common finite support, $D_{\text{KL}}(p\|q)$ denotes Kullback-Leibler divergence in nats and $\text{TV}(p,q) := \frac{1}{2} \sum_i |p_i - q_i|$ the total variation distance. At each block we fix a temperature $\tau_{\text{blk}}$ when mapping similarities to $P^{(t)}$, and we compare only distributions formed with the same $\tau_{\text{blk}}$. All margins $m_t$ are computed on the matched support $\mathcal{I}_t$. The representation vector $u$ is $\ell_2$-normalized and subspaces $U$ have orthonormal columns.

### 3.1 MATCHED SUPPORT FORMULATION

As tokens are progressively unmasked during block-level diffusion (Savinov et al., 2022; Austin et al., 2021), the support of our distribution grows. To compare distributions across steps with different supports, we employ matched-support renormalization. Let $\mathcal{I}_t = \mathcal{S}_{t-1} \cap S_t$ be the intersection of visible tokens between consecutive steps. We renormalize both distributions to this common support:

$$\tilde{P}^{(t)}(s) = \frac{P^{(t)}(s)}{\sum_{i \in \mathcal{I}_t} P^{(t)}(i)}, \qquad \tilde{P}^{(t-1)}(s) = \frac{P^{(t-1)}(s)}{\sum_{i \in \mathcal{I}_t} P^{(t-1)}(i)}, \tag{7}$$

for all $s \in \mathcal{I}_t$. We add $10^{-12}$ smoothing to avoid $\log 0$ in KL computations.

### 3.2 FUNDAMENTAL STABILITY RESULTS

**Lemma 3.1** (Run-length KL implies Multi-step TV Bound). *If the matched-support KL divergences satisfy $D_{t-\Omega+1}, \ldots, D_t \leq \delta$ where $D_t = D_{\mathrm{KL}}(\tilde{P}^{(t)} \| \tilde{P}^{(t-1)})$, then the total variation distance between the endpoints satisfies:*

$$\mathrm{TV}(\tilde{P}^{(t)}, \tilde{P}^{(t-\Omega)}) \leq \Omega\sqrt{\tfrac{\delta}{2}}. \tag{8}$$

*Intuition.* The triangle inequality decomposes multi-step drift into a sum of one-step TV distances, and Pinsker's inequality bounds each term by $\sqrt{D_r/2}$. Summing over the $\Omega$ steps yields the claimed result. A full proof is given in Appendix B.1.

**Theorem 3.2** (Local Argmax Invariance Certificate). *Let $i^*(t) = \arg\max_{s \in \mathcal{I}_t} \tilde{P}^{(t)}(s)$ and $m_t = \tilde{P}^{(t)}_{(1)} - \tilde{P}^{(t)}_{(2)}$ be the top-2 margin on the matched support. If $D_{t-\Omega+1}, \ldots, D_t \leq \delta$ and*

$$\Omega\sqrt{\tfrac{\delta}{2}} < \tfrac{m_t}{2}, \tag{9}$$

*then $i^*(t') = i^*(t)$ for all $t' \in \{t - \Omega, \ldots, t\}$.*

*Intuition.* If the predicted argmax were to change, at least $m_t/2$ probability mass would need to move between the top two coordinates. Lemma 3.1 bounds the multi-step total variation distance by $\Omega\sqrt{\delta/2}$, which is strictly smaller than $m_t/2$ under condition equation 9. Thus, a flip is impossible. A full proof is given in Appendix B.2.

These results show that if RADs remain stable within a small KL window for $\Omega$ consecutive steps, then the predicted token cannot change. This directly justifies the early exit condition implemented in TRACE.

### 3.3 GLOBAL STABILITY UNDER CONTRACTION

**Assumption 3.1** (Post-unmasking Contraction). *After the last unmasking in a block, the Markov operator $K_r$ mapping $\tilde{P}^{(r-1)}$ to $\tilde{P}^{(r)}$ satisfies $\alpha(K_r) \leq \alpha < 1$, where*

$$\alpha(K) := \sup_{p \neq q} \frac{\mathrm{TV}(pK, qK)}{\mathrm{TV}(p, q)} \tag{10}$$

**Theorem 3.3** (Tail Movement Bound and Global Argmax Preservation). *Under Assumption 3.1, if $D_{t-\Omega+1}, \ldots, D_t \leq \delta$, then for any $s \geq 1$:*

$$\mathrm{TV}(\tilde{P}^{(t+s)}, \tilde{P}^{(t)}) \leq \frac{1 - \alpha^s}{1 - \alpha} \cdot \mathrm{TV}(\tilde{P}^{(t)}, \tilde{P}^{(t-1)}) \leq \frac{1}{1 - \alpha}\sqrt{\tfrac{\delta}{2}}. \tag{11}$$

*Consequently, $\sup_{s \geq 1} \mathrm{TV}(\tilde{P}^{(t+s)}, \tilde{P}^{(t)}) \leq \frac{1}{1-\alpha}\sqrt{\tfrac{\delta}{2}}$. If additionally*

$$\Omega\sqrt{\tfrac{\delta}{2}} + \frac{1}{1 - \alpha}\sqrt{\tfrac{\delta}{2}} < \tfrac{m_t}{2}, \tag{12}$$

*then $i^*(t + s) = i^*(t)$ for all $s \geq 0$.*

Table 1: Representative efficiency gains with TRACE early exit (full results in Appendix A). Values show average reduction in denoising steps relative to baseline (64/128/256 steps).

| Dataset | Step Reduction | Accuracy Impact |
|---|---|---|
| GSM8K | 19–33% | ±0.0% |
| MATH500 | 23–41% | ±0.0% |
| Sudoku | 36–42% | +1.5% |
| Countdown | 37–68% | +2.3% |

*Intuition.* The contraction assumption implies one-step deviations shrink by at most a factor $\alpha$. Summing these contributions over $s$ steps gives a geometric bound on the drift. Combining this with Pinsker's inequality and the local margin condition yields global preservation of the argmax. A full proof is provided in Appendix B.3.

**Remark 3.4** (Temperature Robustness)**.** *If the same logits are compared under different temperatures, KL and TV change even when the underlying similarities are identical. We therefore fix $\tau_{blk}$ for all comparisons within a block. As a robustness check, we sweep $\tau_{blk} \in \{0.5, 1.0, 2.0\}$ and report sensitivity curves in Appendix C.*

## 4 EMPIRICAL EVALUATION

Before turning to diagnostics, we briefly recap the motivating efficiency gains. TRACE-style early exits reduce denoising steps by 11–68% across four reasoning benchmarks (GSM8K, MATH500, Sudoku, Countdown) with no degradation of accuracy. Table 1 summarizes representative results (full per-sequence breakdowns are in Appendix A). These observations establish practical feasibility and motivate our central question: *why* are such exits possible, and under what conditions can they be certified?

We analyze both diagnostic association and provable guarantees in this section; formal definitions of correlation ($\rho$), the stability-and-margin certificate, and certified coverage appear in Sec. 4.3.

### 4.1 DATA COLLECTION AND CORE ARTIFACTS

**Readout choice.** We instantiate the RRM readout on the LoRA-$B$ matrix of the final block's Query projection. This is a fixed design choice in this work.

From the AdamW trajectories we form row-$\ell_2$–normalized vectors $u \in \mathbb{R}^d$ (RRMs; Eq. equation 4). At inference step $t$ with visible set $S_t$ of unmasked positions, we compute the RAD $P^{(t)}(s)$ as in Eq. equation 6 from cosine similarities in Eq. equation 5. We assess stability with the matched-support divergence $D_t = \mathrm{KL}\big(\tilde{P}^{(t)} \| \tilde{P}^{(t-1)}\big)$ using the renormalization in Eq. equation 7.

*Storage overhead.* The retained RRM vectors add negligible storage overhead ($\approx 0.02\%$ of an 8 GB model; for QKV across 32 blocks at $d = 4096$ in fp32 this is 1.5 MB, and including $O$ gives 2.0 MB). See App. E for a calculation.

**Setup.** All experiments use sequence length 128, 4 blocks, and 64 denoising steps. For each dataset (GSM8K, MATH500, Sudoku, Countdown) we collect $N=64$ traces and construct an RRM with $d=4096$.

### 4.2 REPRESENTATION METRICS

**Alignment concentration (AC).**

$$\mathrm{AC}^{(t)} = 1 - \frac{H(P^{(t)})}{\log |S_t|}, \qquad H(p) = -\sum_s p_s \log p_s. \tag{13}$$

AC measures how peaked $P^{(t)}$ is relative to uniform over $S_t$ (nats; normalized by $\log |S_t|$).

**Late-step change ratio (LSCR).** Let $D_t$ denote the matched-support divergence between consecutive RAD distributions (we use TV unless noted). We quantify attenuation vs. amplification of step-to-step change via the per-step ratio

$$A_r \ = \ \frac{D_{r+1}}{D_r}, \qquad r \in R_{\text{late}}, \tag{14}$$

and summarize it by the task-level statistic

$$\hat{\alpha}_{\text{task}} \ = \ \max_{r \in R_{\text{late}}} A_r, \tag{15}$$

where $R_{\text{late}}$ indexes late steps after the final unmasking. By construction, $A_r{<}1$ indicates *contraction* (attenuation of change), $A_r{>}1$ indicates *amplification*, and $A_r{\approx}1$ indicates near-neutral change; smaller $\hat{\alpha}_{\text{task}}$ therefore implies stronger contraction. For comparison with the theory, the per-step contraction coefficient there is $\kappa_r = D_{r+1}^{\text{KL}}/D_r^{\text{KL}}$; replacing TV with KL in the LSCR definition leaves the interpretation unchanged.

**Top-2 margin.**

$$m_t \ = \ P_{(1)}^{(t)} - P_{(2)}^{(t)}. \tag{16}$$

### 4.3 DIAGNOSTICS VS. GUARANTEES: DEFINITIONS

**Correlation ($\rho$).** For a per-sample stop step we correlate a representation snapshot (here: AC at stop) with the correctness label. We report point estimates and 95% bootstrap CIs (1,000 resamples).

**Certificate.** Let $\tilde{P}^{(t)}$ be the RAD renormalized to matched support $I_t = S_{t-1} \cap S_t$, and define $D_t = \text{KL}\big(\tilde{P}^{(t)}\big\|\tilde{P}^{(t-1)}\big)$. A $(\delta, \Omega)$ *stability window* holds at stop step $t$ if $D_{t'} \leq \delta$ for every $t' \in \{t - \Omega + 1, \dots, t\}$. By Pinsker and the triangle inequality,

$$\text{TV}\big(\tilde{P}^{(t)}, \tilde{P}^{(t-k)}\big) \ \leq \ k\sqrt{\delta/2} \quad \text{for } k \leq \Omega,$$

so the whole window lies inside a TV ball of radius $r_t := \Omega\sqrt{\delta/2}$ around $\tilde{P}^{(t)}$. Let $m_t := p_{(1)}^{(t)} - p_{(2)}^{(t)}$ denote the top-2 probability margin under $P^{(t)}$. If $m_t > 2\,r_t$, then the argmax at step $t$ cannot flip anywhere within the window (no-flip). A sample satisfying both conditions is *certified.*

**Coverage.** For a dataset, coverage is the percentage of certified samples under a specified readout (strict or calibrated) and $(\delta, \Omega)$.

**Hyperparameter selection (grid over $\delta$ and $\Omega$).** We select certificate parameters by sweeping $\delta$ and $\Omega$ on a held-out validation split and then *fix* a single pair for all experiments. Concretely, we grid $\delta \in \{10^{-6}, 3{\times}10^{-6}, 10^{-5}\}$ and $\Omega \in \{1, 2, 3, 4\}$ at late steps ($L{=}24$ after the final unmasking), choose $(\delta^\star, \Omega^\star)$ using the $Q$ readout, and reuse it for all projections and aggregates. In our runs the stability window was the bottleneck, so certified coverage *increased monotonically* as $\delta$ relaxed from $10^{-6}$ to $10^{-5}$ at fixed $\Omega{=}3$, while the margin remained sufficient—motivating $(\delta^\star, \Omega^\star){=}(3{\times}10^{-6}, 3)$ for all results in §4.

**Projection ablation (Q/K/V and simple aggregates).** We compared readouts on the final block's $\{Q, K, V\}$ projections and two aggregates (QKV-mean/max), holding $(\delta, \Omega) = (3 \times 10^{-6}, 3)$ fixed at late steps ($L{=}24$). Results (App. Table 4) show: (i) the strongest single projection is *dataset-dependent*—$Q$ wins on GSM8K and Sudoku, $K$ on MATH500, and $V$ dominates Countdown; (ii) QKV-mean never exceeds the winner, matching the best on Sudoku, equaling $Q$ on MATH500, but substantially degrading coverage on Countdown when one branch ($V$) carries most of the signal—consistent with averaging shrinking the top-2 margin once the KL window is already satisfied; and (iii) QKV-max tracks the best branch on GSM8K and is close on Countdown, but lags on MATH500/Sudoku, indicating that tokenwise maximization does not necessarily preserve the stop-step margin. Our LoRA configuration attaches adapters to $Q/K/V$ only at the final block (no LoRA on the output projection), so $O$/QKVO are omitted. These observations support our choice to report the $Q$ readout in the main results: it is strong on two of four tasks, simple to deploy, and avoids aggregate-induced margin shrinkage.

### 4.4 Empirical Profiles at seq=128

Global summaries (means/medians) for the four datasets show:

- **AC (final)**: small and tightly clustered ($\sim 10^{-5}$). Mean AC is slightly higher for MATH500/Countdown ($\approx 1.8 \times 10^{-5}$) than Sudoku ($\approx 1.35 \times 10^{-5}$); GSM8K is intermediate.
- **Late-step change ratio**: Sudoku exhibits the smallest late-step amplification (mean $\hat{\alpha} \approx 2.08$), while Countdown shows the largest ($\approx 2.56$); GSM8K/MATH500 lie in between ($\approx 2.39/2.52$). Note that in this slice $\hat{\alpha} > 1$ indicates amplification rather than contraction.
- **Margins (final)**: GSM8K/Countdown exhibit larger margins ($\sim 1.5$–$1.7 \times 10^{-4}$) than MATH500/Sudoku ($\sim 0.9$–$1.4 \times 10^{-4}$).

These dynamics motivate certification analysis: margins and late-step stabilization jointly matter.

### 4.5 Certified Early Stops: Strict vs. Calibrated

**Strict (baseline) certificate.** With $P^{(t)}$ built at $\tau_{\text{blk}} = 1.0$ and Pinsker's sufficient bound, multi-step stability with $\Omega \geq 2$ produces negligible coverage across tasks: final-step margins ($\sim 10^{-4}$) are too small relative to the implied TV balls from observed KLs. If we relax to a *single-step window* ($\Omega = 1$), the bound becomes attainable on a sizable fraction of samples (Table 2).

**Late window.** Let $T$ denote the total number of denoising steps. Unless otherwise noted, we evaluate certification over the last $L=24$ steps *after the final unmasking* (the *late window*): $R_{\text{late}} := \{T-L+1, \ldots, T\}$. In this region the visible set is constant, i.e., $S_t = S_T$ for all $t \in R_{\text{late}}$. All reported coverage and LSCR statistics are computed on $R_{\text{late}}$.

**Calibrated certificate.** Following §4.3, we evaluate a monotone readout calibration that preserves matched support. If per-step similarities $\{\text{Sim}^{(t)}\}$ are logged, we rebuild the distribution at a lower temperature $\tau < 1$:

$$\hat{P}^{(t)}(s) = \frac{\exp\big(\text{Sim}_s^{(t)}/\tau\big)}{\sum_{i \in S_t} \exp\big(\text{Sim}_i^{(t)}/\tau\big)}.$$

Otherwise we apply power sharpening to the existing distribution:

$$\hat{P}^{(t)}(s) = \frac{\big(P^{(t)}(s)\big)^{\gamma}}{\sum_{i \in S_t} \big(P^{(t)}(i)\big)^{\gamma}}.$$

The *strict* variant corresponds to $\tau=1$ (or $\gamma=1$). Calibration can increase the top-2 margin but may also increase late-window KL; its net effect on certified coverage is therefore data-dependent. In our slice, calibrated coverage is *lower* than strict across all four datasets (Table 2).

### 4.6 Results: Correlation and Coverage

Table 2 reports *diagnostic association* (Spearman $\rho$ of AC-at-stop vs. accuracy) and *provable coverage* (strict $\tau=1.0$ vs. calibrated settings). Sudoku correlations use *cell-wise accuracy*.

**Interpretation.** First, $\rho$ is consistently weak—unsurprising for a *single-step* snapshot of a multi-step trajectory. Second, the certificate is where safety enters: under the *same* $(\delta, \Omega)$ rule and a late window of 24 (see below), *strict* readouts ($\tau=1.0$) already certify a substantial fraction of samples (notably Countdown and GSM8K) at $\Omega=1$. In this particular slice, the *calibrated* readouts ($\tau=0.5$ or $P^{\gamma}$, $\gamma=1.5$) yield lower coverage than strict across all four datasets. This is consistent with cases where sharpening increases late-step KL more than it increases the margin, making the no-flip inequality harder to satisfy at the chosen operating point. The certificate itself remains explicit and auditable: per sample we test both the KL window and the margin inequality, and coverage is the fraction that pass.

| Dataset | $\rho$ (95% CI) | Strict coverage | Calibrated coverage |
|---|---|---|---|
| GSM8K | +0.069 (-0.160, +0.295) | 62.50% | 14.06% |
| MATH500 | -0.149 (-0.384, +0.103) | 59.38% | 6.25% |
| Sudoku | -0.184 (-0.423, +0.065) | 23.44% | 1.56% |
| Countdown | +0.136 (-0.121, +0.363) | 75.00% | 6.25% |

Table 2: **Diagnostics vs. guarantees at seq=128 (N=64 per dataset).** Spearman correlation of AC at stop vs. accuracy (95% bootstrap CI), and certified coverage under *strict* ($\tau$=1.0) and *calibrated* readouts at fixed parameters ($\delta$=$10^{-8}$, $\Omega$=1) and the late window ($L$=24).

**Takeaways.** (i) A diagnostic snapshot (AC-at-stop) is not strongly predictive on its own; (ii) provable certificates give a safety lens that can meaningfully gate early stopping; (iii) the practical coverage you get is a function of both the stability span $\Omega$ and the readout (strict vs. calibrated). We therefore advocate reporting *both* views side-by-side.

## 5 RELATED WORK

Concurrent algorithmic work has focused on demonstrating the feasibility of metadata-guided early exits in diffusion language models, reporting substantial efficiency gains in practice. By contrast, our contribution in TRACE is to provide the theoretical account: we formalize how training dynamics induce actionable representations, prove stability guarantees, and analyze their structure across reasoning tasks.

### 5.1 TRAINING DYNAMICS AND REPRESENTATION LEARNING

The relationship between optimization trajectories and learned representations has been studied from multiple perspectives. Frankle & Carbin (2019) demonstrated that specific parameter subsets emerge as critical during training, while Fort & Ganguli (2019) showed that neural networks learn hierarchical representations progressively. Our work extends these insights by preserving optimization metadata for inference-time use.

### 5.2 ADAPTIVE COMPUTATION IN NEURAL NETWORKS

Adaptive computation has a rich history in neural networks. Graves (2016) introduced adaptive computation time for RNNs, allowing models to dynamically allocate processing steps. Figurnov et al. (2017) extended this to spatial dimensions in CNNs. For transformers, Schwartz et al. (2020) proposed depth-adaptive inference, while Schuster et al. (2022) developed confidence-based early exiting. TRACE builds upon this foundation by leveraging training-time information rather than relying solely on inference-time signals.

### 5.3 DIFFUSION MODELS FOR LANGUAGE

Diffusion models have recently been adapted for discrete text generation. Austin et al. (2021) introduced discrete diffusion with absorbing states, while Savinov et al. (2022) proposed step-unrolled denoising autoencoders. Dieleman et al. (2022) developed continuous diffusion for categorical data. Our contribution is orthogonal to these architectural advances, focusing on leveraging training dynamics for efficient inference regardless of the specific diffusion formulation.

### 5.4 PRACTICAL DEPLOYMENT AND EFFICIENT INFERENCE

Beyond algorithm design, there is growing interest in systems for efficient large language model (LLM) deployment. Recent work explores speculative decoding Leviathan et al. (2023), dynamic batching and scheduling Narayanan et al. (2021), and mixture-of-experts routing Du et al. (2022) as ways to reduce inference cost, while early-exit methods have been studied in autoregressive transformers Liu et al. (2023); Elhage et al. (2021). These

approaches rely primarily on inference-time heuristics such as confidence thresholds or calibration. TRACE offers a complementary perspective: by leveraging training-time metadata, it provides a principled signal for adaptive termination that can be certified for safety. This suggests a broader deployment paradigm where training dynamics are preserved and exposed as lightweight metadata, enabling inference systems to allocate compute adaptively without additional fine-tuning or architectural changes.

## 6 DISCUSSION

Our analysis establishes that training dynamics not only yield efficiency gains but also admit formal guarantees: TRACE explains *why* adaptive early exits are safe. By preserving optimizer trajectories, we show that training and inference are linked through representational stability rather than being disjoint phases.

**Practical implications.** Current deployment pipelines discard optimizer metadata, yet our results show this information can be retained at negligible cost ($\sim 0.02\%$ for an 8GB model; App. E) and reused for inference-time decisions. This suggests a broader paradigm where compute allocation is guided by certified stability conditions instead of fixed budgets.

**Theory–empirics alignment.** Experiments confirm that matched-support KL→TV bounds predict inference behavior: once RAD stability windows are satisfied, the predicted argmax remains invariant. Certificates thus act as auditable safety checks, bridging formal guarantees with observed efficiency.

**Limitations.** TRACE assumes access to optimizer states from fine-tuning, focuses on LoRA-B(WQ) modules, and is instantiated for block-level diffusion. Extending to other adaptation methods, full-parameter fine-tuning, or autoregressive transformers remains open.

**Implications for reasoning.** On mathematical tasks, RAD stabilization reflects problem completion, making certified early termination a principled efficiency lever. Diagnostics such as alignment concentration and the late-step change ratio provide model-agnostic probes of reasoning stability. Similar fingerprints may emerge in domains like logic, program synthesis, or multimodal reasoning.

In sum, TRACE highlights that optimization dynamics encode actionable, certifiable signals. Treating optimizer states as first-class objects enables inference that is both more efficient and more interpretable, grounded in the model's own training history.

## 7 CONCLUSION

We introduced TRACE, a framework that formalizes *Reasoning Representation Maps* (RRMs) from optimizer trajectories and defines *Representational Alignment Distributions* (RADs) for inference. By proving KL→TV stability certificates, we established conditions under which predictions remain invariant, providing a principled explanation of why adaptive early exits are safe. TRACE implements these ideas in practice, consistently reducing denoising steps without accuracy loss.

Beyond efficiency, our results highlight a broader principle: optimizer states, often discarded, encode actionable fingerprints of learned capabilities. Preserving and exposing this metadata enables inference that is both adaptive and certifiable. While our work assumes access to fine-tuning trajectories and is instantiated on block-level diffusion with LoRA adapters, the approach suggests a general perspective where training and inference are connected through representational stability.

Future directions include extending TRACE to autoregressive transformers, exploring domains such as program synthesis or multimodal reasoning, and standardizing metadata retention for large-scale deployment. Together, these directions point toward a more holistic view of machine learning pipelines, where optimization dynamics are treated as first-class objects and leveraged for efficient, interpretable, and safe inference.

**Ethics Statement.** We have read and will adhere to the ICLR Code of Ethics. This work does not involve human subjects, sensitive attributes, or personally identifiable information. All datasets are publicly available and used under their respective licenses; we document licenses and any filtering in the appendix. We discuss potential risks (e.g., misuse of efficiency techniques, distributional bias) and mitigation steps in Appendix § [Ethics/Limitations]. We are not aware of legal, safety, or privacy issues specific to this study.

**Reproducibility Statement.** We provide the exact steps to reproduce all results. The appendix details datasets, preprocessing, and artifact hashes; training and evaluation commands; hyperparameters; and hardware settings. After corporate legal approval, we plan to open source the supplementary materials that include code archive with environment specification, scripts to collect optimizer trajectories, construct RRM/RAD, run TRACE, and reproduce all tables/figures from a fresh checkout. We will also include seeds and configuration files for every experiment.

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

# A  Motivating Early Exit Results

To complement the main text, we provide a compact summary of the TRACE early exit procedure and representative empirical benefits observed on standard reasoning benchmarks. These results are included to illustrate the feasibility of metadata-guided early termination; the primary contribution of this paper remains the theoretical analysis of representation stability.

## A.1  TRACE Early Exit Algorithm

---

**Algorithm 1** Training-Refined Adaptive Computation Exit (TRACE) Uses Reasoning Representation Maps and Representational Alignment Distributions (RADs) to decide early exit.

---

1: **Input:** Input sequence; threshold $\delta$; span $\Omega$; block temperature $\tau_{\text{blk}}$; reasoning vector $u$
2: **Output:** Generated sequence with adaptive early termination

3: Initialize stability counter $c \leftarrow 0$
4: For each diffusion block $b = 1, \ldots, B$:
5:    Initialize visible token set $S_1$
6:    Compute activations $f_s^{(1)}$ for $s \in S_1$
7:    Form alignment distribution $P^{(1)}$ (Eq. equation 6)
8:    Set $c \leftarrow 0$
9:    For each denoising step $t = 2, \ldots, T_b$:
10:       Update visible set $S_t$
11:       Compute activations $f_s^{(t)}$ and distribution $P^{(t)}$
12:       Restrict to $I_t = S_{t-1} \cap S_t$; renormalize to $\tilde{P}^{(t)}$, $\tilde{P}^{(t-1)}$
13:       Compute $D_t = \text{KL}(\tilde{P}^{(t)} \parallel \tilde{P}^{(t-1)})$
14:       If $D_t < \delta$: set $c \leftarrow c + 1$; else set $c \leftarrow 0$
15:       If $c \geq \Omega$: break (early exit for block $b$)

---

## A.2  Representative Efficiency Gains

Table 3 reports the full per-sequence results underlying the summary in Table 1. Baseline runs use fixed denoising steps (64/128/256 for sequence lengths 128/256/512). TRACE adaptively halts earlier once representational stability is detected, reducing computation by 11–68% with accuracy maintained.

Table 3: Full efficiency gains with TRACE early exit across four reasoning datasets. Values are averaged across blocks; percentages indicate reduction relative to baseline diffusion steps.

| Dataset | Seq=128 | Seq=256 | Seq=512 | Accuracy Impact |
|---|---|---|---|---|
| GSM8K | 33% fewer steps | 19% fewer steps | 12% fewer steps | ±0.0% |
| MATH500 | 41% fewer steps | 36% fewer steps | 23% fewer steps | ±0.0% |
| Sudoku | 40% fewer steps | 42% fewer steps | 36% fewer steps | +1.5% |
| Countdown | 37% fewer steps | 68% fewer steps | 48% fewer steps | +2.3% |

**Takeaway.** TRACE consistently reduces denoising steps with negligible or positive effects on accuracy. These full results complement the main-text summary in Table 1.

## B   EXTENDED THEORETICAL RESULTS

Throughout, $\Delta$ denotes the probability simplex on the matched support $\mathcal{I}_t$.

### B.1   PROOF OF LEMMA 3.1

*Proof of Lemma 3.1.* By the triangle inequality for total variation,

$$\mathrm{TV}(\tilde{P}^{(t)}, \tilde{P}^{(t-\Omega)}) \;\leq\; \sum_{r=t-\Omega+1}^{t} \mathrm{TV}(\tilde{P}^{(r)}, \tilde{P}^{(r-1)}).$$

Applying Pinsker's inequality $\mathrm{TV}(p,q) \leq \sqrt{D_{\mathrm{KL}}(p\|q)/2}$ to each summand and using $D_r \leq \delta$ for $r = t - \Omega + 1, \ldots, t$,

$$\mathrm{TV}(\tilde{P}^{(t)}, \tilde{P}^{(t-\Omega)}) \;\leq\; \sum_{r=t-\Omega+1}^{t} \sqrt{\frac{D_r}{2}} \;\leq\; \Omega\sqrt{\frac{\delta}{2}},$$

which establishes equation 8. $\qquad\square$

### B.2   PROOF OF THEOREM 3.2

*Proof of Theorem 3.2.* Suppose the argmax changes between $\tilde{P}^{(t)}$ and some $\tilde{P}^{(t')}$ with $t' \in \{t-\Omega, \ldots, t\}$. For the top element to change position, at least $m_t/2$ probability mass must move between the top two coordinates. This implies $\mathrm{TV}(\tilde{P}^{(t)}, \tilde{P}^{(t')}) \geq m_t/2$. However, Lemma 3.1 bounds this distance by $\Omega\sqrt{\delta/2} < m_t/2$, yielding a contradiction. Therefore, the argmax remains invariant across the specified window. $\qquad\square$

### B.3   PROOF OF THEOREM 3.3

*Proof of Theorem 3.3.* The one-step contraction property gives

$$\mathrm{TV}(\tilde{P}^{(r+1)}, \tilde{P}^{(r)}) \leq \alpha \cdot \mathrm{TV}(\tilde{P}^{(r)}, \tilde{P}^{(r-1)}).$$

Iterating $s$ times yields

$$\mathrm{TV}(\tilde{P}^{(t+j)}, \tilde{P}^{(t+j-1)}) \leq \alpha^j \cdot \mathrm{TV}(\tilde{P}^{(t)}, \tilde{P}^{(t-1)}) \quad \text{for } j = 1, \ldots, s.$$

Summing the geometric series gives

$$\mathrm{TV}(\tilde{P}^{(t+s)}, \tilde{P}^{(t)}) \leq \sum_{j=0}^{s-1} \alpha^j \cdot \mathrm{TV}(\tilde{P}^{(t)}, \tilde{P}^{(t-1)}) = \frac{1-\alpha^s}{1-\alpha} \, \mathrm{TV}(\tilde{P}^{(t)}, \tilde{P}^{(t-1)}).$$

Applying Pinsker's inequality and invoking the margin argument from Theorem 3.2 establishes the bound in equation 11 and the preservation condition in equation 12. $\qquad\square$

### B.4 Stability of Lipschitz Observables

**Theorem B.1** (Stability of Lipschitz Functionals)**.** *Let $F : \Delta \to \mathbb{R}$ be $L$–Lipschitz in total variation, i.e., $|F(p) - F(q)| \leq L \cdot \mathrm{TV}(p, q)$ for all $p, q \in \Delta$. If $D_{t-\Omega+1}, \ldots, D_t \leq \delta$, then*

$$\left| F(\tilde{P}^{(t)}) - F(\tilde{P}^{(t-\Omega)}) \right| \leq L \cdot \Omega \sqrt{\frac{\delta}{2}}. \tag{17}$$

*Under Assumption 3.1,*

$$\sup_{s \geq 1} \left| F(\tilde{P}^{(t+s)}) - F(\tilde{P}^{(t)}) \right| \leq \frac{L}{1-\alpha} \sqrt{\frac{\delta}{2}}. \tag{18}$$

*Proof.* Apply Lemma 3.1 and Theorem 3.3 and use the Lipschitz property of $F$ in TV. $\square$

### B.5 Practical Calibration Guarantees

**Corollary B.2** (PAC-style No-Flip Calibration)**.** *Let $M$ denote the top-2 margin at TRACE's stopping time on a validation set, and let $q_{1-\beta}$ be the $(1 - \beta)$-quantile of $M$. Estimate a post-unmasking contraction bound by*

$$\hat{\alpha} = \max_{\text{val instances, late } r} \frac{\mathrm{TV}(\tilde{P}^{(r+1)}, \tilde{P}^{(r)})}{\mathrm{TV}(\tilde{P}^{(r)}, \tilde{P}^{(r-1)})}. \tag{19}$$

*Choose $(\delta, \Omega)$ such that*

$$\Omega \sqrt{\frac{\delta}{2}} + \frac{1}{1-\hat{\alpha}} \sqrt{\frac{\delta}{2}} \leq \frac{1}{2} q_{1-\beta}. \tag{20}$$

*Then, with probability at least $1 - \beta$ over test instances, the top-1 token at TRACE's stop equals the top-1 token under continued denoising on the fixed support.*

*Proof.* Combine Theorem 3.3 with the margin condition of Theorem 3.2. $\square$

## C Temperature Sensitivity

The temperature $\tau_{\mathrm{blk}}$ sets the sharpness of the Representational Alignment Distribution defined in §2.2. For robustness we fix $\tau_{\mathrm{blk}}$ per block when comparing distributions and assess sensitivity by sweeping $\tau_{\mathrm{blk}} \in \{0.5, 1.0, 2.0\}$ on validation data, reporting any changes in stability windows $D_t$ and margins $m_t$.

## D Bias Correction Equivalence

We empirically compare Reasoning Representation Maps computed from bias-corrected vs. uncorrected AdamW moments (see §2.1) via (i) vector correlations for $u$ and (ii) principal angles for subspace variants $U$. We report agreement statistics and any downstream differences in RAD-based metrics.

## E Storage Overhead and Retention Policy

At deployment we retain only the RRM vectors and a few calibration scalars. If $|\mathcal{P}|$ modules are used, each with RRM dimension $d$ and $b$ bytes per entry (2 for fp16, 4 for fp32), the storage is

$$\text{bytes}_{\mathrm{RRM}} = |\mathcal{P}| \cdot d \cdot b, \qquad \text{overhead}_\% = 100 \times \frac{\text{bytes}_{\mathrm{RRM}}}{\text{model bytes}}.$$

*Examples ( $d = 4096$, model $\approx 8\,\mathrm{GiB}$):*

- $|\mathcal{P}| = 1$, fp16 $\Rightarrow$ 8,192 bytes (8 KiB) $\Rightarrow$ 0.000095%.

- $|\mathcal{P}| = 16$ (4 blocks $\times$ {Q,K,V,O}), fp16 $\Rightarrow$ 131,072 bytes (128 KiB) $\Rightarrow$ 0.00153%.

- $|\mathcal{P}| = 96$ (QKV across 32 blocks), fp32 $\Rightarrow$ 1,572,864 bytes (1.5 MB) $\Rightarrow$ 0.018%.

- $|\mathcal{P}| = 128$ (QKVO across 32 blocks), fp32 $\Rightarrow$ 2,097,152 bytes (2.0 MB) $\Rightarrow$ 0.024%.

Training-time logs used to construct RRMs are discarded after aggregation.

## F  REPRODUCIBILITY: END-TO-END PIPELINE

This section provides a numbered, end-to-end recipe to reproduce all TRACE results. Each step specifies: (i) the script invoked, (ii) the exact command, and (iii) what the script does, with a reference to its algorithm box.

### STEP 1. COLLECT CORE METRICS

**Script.** `trace_collect_trace_core_metrics.py` constructs the reasoning vector $u$ from cached AdamW states, attaches hooks for activations, and logs per-step similarities, RADs, and matched-support KLs. See Algorithm 2 in Figure 1.

**Command.**

```
CUDA_VISIBLE_DEVICES=0 python trace_collect_trace_core_metrics.py \
    --model_path GSAI-ML/LLaDA-8B-Instruct \
    --dataset gsm8k \
    --gen_length 128 --block_length 32 --diffusion_steps 64 \
    --subsample 64 --output_dir . \
    --checkpoint_path ./checkpoints/SFT/checkpoint-200
```

---

**Algorithm 2: TRACE Core Metric Collector** (`trace_collect_trace_core_metrics_ext.py`).
Builds the reasoning vector and logs per-step traces.

**Inputs.** Dataset identifier; prompts JSONL; model path; LoRA-derived $u$; block index; number of steps; $\tau_{\mathrm{blk}}$.

**Outputs.** Per-sample JSON trace with visible positions, $\mathrm{Sim}^{(t)}$, $P^{(t)}$, and KL divergences.

**Procedure.**

1. Load model and tokenizer; install hook on $q$-projection.
2. Load and normalize $u$.
3. For each prompt:
   (a) Generate masked variants for denoising.
   (b) Capture activations $f_s^{(t)}$.
   (c) Compute similarities and $P^{(t)}$.
   (d) Renormalize and compute KL divergence.
   (e) Append step record.
4. Save JSON trace.

---

Figure 1: **TRACE Core Metric Collector.** Implements artifact collection described in Appendix F.

### STEP 2. DIGEST CORE METRICS

**Script.** `trace_digest_core_metrics.py` aggregates per-sample traces into CSV/JSON summaries, verifies $\|u\|_2 = 1$, and computes global statistics (means/medians of $D_t$, entropies, token counts). See Algorithm 3 in Figure 2.

**Command.**

```
python trace_digest_core_metrics.py \
  ——trace_dir trace_core \
  ——dataset gsm8k \
  ——out_dir logs
```

---

**Algorithm 3: Digest Core Metrics (`trace_digest_core_metrics.py`).**
Aggregates per-sample traces into global summaries.

**Inputs.** Directory of per-sample JSON traces.
**Outputs.** Per-sample CSVs and a global JSON with summary statistics.

**Procedure.**

1. Load and validate traces; check $\|u\|_2 = 1$.

2. Compute averages of $D_t$, entropies, visible token counts.

3. Write per-sample CSV and global JSON summary.

---

Figure 2: **Digest Core Metrics.** Aggregates JSON traces into summary CSV/JSON files.

STEP 3. ANALYZE TASK METRICS

**Script.** `trace_analyze_task_metrics.py` computes alignment concentration (AC), the LSCR statistic $\hat{\alpha}_{\text{task}}$ (Eq. equation 15), top-2 margins, and per-sample certification flags.

See Algorithm 4 in Figure 3.

**Command.**

```
python trace_analyze_task_metrics.py \
  ——trace_dir trace_core \
  ——dataset sudoku \
  ——out_dir logs
```

---

**Algorithm 4: Task Metrics & Certification (`trace_analyze_task_metrics.py`).**
Computes §4.2 metrics from TRACE artifacts and emits per-sample and global summaries.

**Inputs.** TRACE JSON traces.
**Outputs.** Per-sample CSV with AC, margins, certification flags; global JSON summary.

**Procedure.**

1. Load traces.

2. Compute AC trajectories and final AC.

3. Compute late-step change ratio (LSCR) $\hat{\alpha}_{\text{task}}$ (Eq. equation 15).

4. Compute top-2 margins $m_t$.

5. Evaluate certification rule (KL window + Pinsker margin).

6. Write per-sample CSV and global JSON.

---

Figure 3: **Task Metrics & Certification.** Implements AC, LSCR (late-step change ratio; Eq. equation 15), margins, and certification flags.

STEP 4. DIRECT CERTIFICATION

**Script.** `trace_certified_stops.py` tests a fixed $(\delta, \Omega)$ certificate using matched-support KL windows and the no-flip margin condition. See Algorithm 5 in Figure 4.

**Command.**

```
python trace_certified_stops.py \
  ——trace_dir trace_core \
```

```
810    ——dataset gsm8k \
811    ——out_dir logs \
812    ——delta 1e−8 ——omega 3
```

---

**Algorithm 5: Certified Stopping (`trace_certified_stops.py`).**
Evaluates the certificate at fixed $(\delta, \Omega)$.

**Inputs.** TRACE JSON traces; $\delta$, $\Omega$.
**Outputs.** Per-sample CSV with certification flags and $t^\star$; global JSON with coverage.

**Procedure.**

1. Load per-sample traces.

2. Normalize distributions on visible sets.

3. Compute matched-support KL $D_t$ and margins $m_t$.

4. Apply stability + no-flip test in late window.

5. Write CSV and global JSON coverage.

---

Figure 4: **Certified Stopping.** Applies $(\delta, \Omega)$ rule to evaluate certified stops.

STEP 5. CERTIFICATION SWEEPS

**Scripts.**

- `trace_certified_from_sim.py`: rebuilds $P^{(t)}$ from stored Sim at multiple $\tau$ values. See Algorithm 6.

- `trace_certified_temp_sweep.py`: sharpens stored $P^{(t)}$ via $P^\gamma$ and sweeps $(\gamma, \delta, \Omega)$. See Algorithm 7.

**Command (from similarities).**

```
python trace_certified_from_sim.py \
    ——trace_dir trace_core \
    ——dataset math \
    ——out_dir logs \
    ——taus "1,0.75,0.5,0.35" \
    ——deltas "1e−6,1e−7,1e−8" \
    ——omegas "1,2,3" ——late_window 24
```

**Command (power sharpening).**

```
python trace_certified_temp_sweep.py \
    ——trace_dir trace_core \
    ——dataset countdown \
    ——out_dir logs \
    ——gammas "1.5,2,3" \
    ——deltas "1e−8,1e−9" \
    ——omegas "1,2,3" ——late_window 24
```

STEP 6. MERGE LABELS AND DIAGNOSTICS

**Script.** `trace_sec44_extract_csv.py` merges per-sample labels with TRACE metrics and computes $\rho$, CIs, and certified coverage. See Algorithm 8.

**Command.**

```
python trace_sec44_extract_csv.py \
    ——json_dir ./json \
    ——out_dir out_trace \
    ——trace_metrics_dir logs
```

---

**Algorithm 6: Certification Sweeps from Similarities (`trace_certified_from_sim.py`).**
Recomputes $P^{(t)}$ at alternative $\tau$ and applies the matched-support certificate.

**Inputs.** TRACE JSONs with Sim; list of $\tau, \delta, \Omega$.
**Outputs.** Coverage CSVs/JSONs for grid of $(\tau, \delta, \Omega)$.

**Procedure.**

1. Load per-sample traces with similarities.

2. For each $\tau$, recompute RADs $P_\tau^{(t)}$.

3. Compute matched-support KL $D_t$ and margins $m_t$.

4. Apply certificate across $(\delta, \Omega)$.

5. Write coverage grid CSV/JSON.

Figure 5: **Certification sweeps (from Sim).**

---

**Algorithm 7: Certification Sweeps via Power Sharpening (`trace_certified_temp_sweep.py`).**
Sharpens stored $P^{(t)}$ via $P^\gamma$ and applies the matched-support certificate.

**Inputs.** TRACE JSONs with $P^{(t)}$; list of $\gamma, \delta, \Omega$.
**Outputs.** Coverage CSVs/JSONs for grid of $(\gamma, \delta, \Omega)$.

**Procedure.**

1. Load per-sample traces with stored $P^{(t)}$.

2. For each $\gamma$, sharpen $P^{(t)}$ and renormalize.

3. Compute matched-support KL $D_t$ and margins $m_t$.

4. Apply certificate across $(\delta, \Omega)$.

5. Write coverage grid CSV/JSON.

Figure 6: **Certification sweeps (power sharpening).**

---

**Algorithm 8: Labels and Diagnostics (`trace_sec44_extract_csv.py`).**
Merges ground-truth labels with TRACE metrics and computes §4.4 diagnostics.

**Inputs.** JSON labels; optional TRACE metrics CSVs.
**Outputs.** Labels CSVs; merged CSV; sec44_stats.csv.

**Procedure.**

1. Load and normalize labels.

2. Merge with TRACE metrics if provided.

3. Compute per-dataset accuracy summaries.

4. Compute $\rho$, bootstrap CI, and certified coverage.

5. Write results to CSV/JSON.

Figure 7: **Merge Labels and Diagnostics.**

STEP 7. DUAL STRICT VS. CALIBRATED CERTIFICATION

**Script.** `trace_sec43_certify_dual_v2.py` compares strict ($\tau=1.0$) vs. calibrated coverage. See Algorithm 9.

**Command.**

```
python trace_sec43_certify_dual_v2.py \
  --trace_dir trace_core \
  --dataset gsm8k \
  --out_dir logs \
  --delta 1e-8 --omega 1 \
  --tau_calib 0.50 --gamma_calib 1.5 \
  --late_window 24
```

---

**Algorithm 9: Dual Strict vs. Calibrated Certification (`trace_sec43_certify_dual_v2.py`).**
Compares strict vs. calibrated coverage under the same certificate.

**Inputs.** TRACE JSONs; $\delta, \Omega$; calibration params $\tau, \gamma$.
**Outputs.** Per-sample CSV and global JSON with strict vs. calibrated coverage.

**Procedure.**

1. Build strict distributions from stored $P^{(t)}$.

2. Build calibrated distributions from Sim or $P^{\gamma}$.

3. Compute matched-support KL and margins.

4. Apply stability + no-flip certificate.

5. Write strict and calibrated results to CSV/JSON.

---

Figure 8: **Dual Certification.** Strict vs. calibrated coverage side-by-side.

STEP 8. BUILD FINAL TABLE

**Script.** `trace_sec44_make_table.py` generates the LaTeX table and prose. See Algorithm 10.

**Command.**

```
python trace_sec44_make_table.py
```

---

**Algorithm 10: Table Emitter (`trace_sec44_make_table.py`).**
Assembles the §4.4 summary table and prose from diagnostics and coverage JSONs.

**Inputs.** sec44_stats.csv; dual_cert_v2_{dataset}_global.json.
**Outputs.** sec44_table.tex; sec44_table_preview.txt; sec44_paragraph.txt.

**Procedure.**

1. Load diagnostics from sec44_stats.csv.

2. Load coverage from JSONs.

3. Write preview text table.

4. Write LaTeX table.

5. Write prose summary.

---

Figure 9: **Final Table Emitter.** Generates table and prose for §4.4.

# G  CERTIFICATE HYPERPARAMETER SWEEP AND SELECTION

**Protocol.** We sweep $\delta \in \{10^{-6}, 3 \times 10^{-6}, 10^{-5}\}$ and $\Omega \in \{1, 2, 3, 4\}$ at late steps ($L = 24$ after the final unmasking) on a held-out validation split. We select $(\delta^{\star}, \Omega^{\star})$ using the $Q$ readout only and then *hold it fixed* for all projections (Q/K/V) and aggregates (QKV-mean/max) to ensure fair comparisons.

**Observed trends.** Across datasets, coverage increases as $\delta$ relaxes in this range (the stability window ceases to be the limiter) while the top-2 margin remains large enough to satisfy the no-flip inequality. Coverage trades off predictably with $\Omega$ (larger windows are stricter). We therefore set $(\delta^{\star}, \Omega^{\star}) = (3 \times 10^{-6}, 3)$ for all main-text experiments.

**Reproducible commands.**

```
# Grid over delta x Omega at late steps L=24 (strict, Q readout)
python trace_certified_grid.py \
  --trace_dir traces_ablate/gsm8k_Q \
```

```
--dataset gsm8k --out_dir logs/gsm8k_Q \
--deltas "1e-6,3e-6,1e-5" --omegas "1,2,3,4" \
--late_window 24 | tee logs/cert_grid_gsm8k_Q.log

# Repeat for: math/sudoku/countdown (and optionally for K/V if desired)
```

**Optional calibration (single scalar tuned on validation).** When per-step similarities are logged, we evaluate a temperature calibration that preserves matched support by rebuilding $P^{(t)}$ at lower $\tau$; when only probabilities are stored, we use power sharpening $P^{(t)} \mapsto P^{(t)\gamma}/\|P^{(t)\gamma}\|_1$. We tune $\tau$ (or $\gamma$) on the same validation split at fixed $(\delta^\star, \Omega^\star)$ and keep it constant across rows.

```
# Temperature sweep (uses stored similarities; strict is tau=1.0)
python trace_certified_from_sim.py \
  --trace_dir traces_ablate/gsm8k_Q \
  --dataset gsm8k --out_dir logs/gsm8k_Q \
  --taus "1.0,0.75,0.5,0.35" \
  --deltas "3e-6" --omegas "3" --late_window 24 \
  | tee logs/cert_tau_gsm8k_Q.log

# Power sharpening sweep (operates on stored probabilities)
python trace_certified_temp_sweep.py \
  --trace_dir traces_ablate/gsm8k_Q \
  --dataset gsm8k --out_dir logs/gsm8k_Q \
  --gammas "1.25,1.5,2,3" \
  --deltas "3e-6" --omegas "3" --late_window 24 \
  | tee logs/cert_gamma_gsm8k_Q.log
```

**Selection rule.** We pick $(\delta^\star, \Omega^\star)$ to maximize certified coverage on the validation split under the $Q$ readout, then reuse it for all projections and aggregates. If calibration is reported, we select a single $\tau^\star$ (or $\gamma^\star$) on validation at $(\delta^\star, \Omega^\star)$ and apply it uniformly.

Table 4: **Projection ablation at fixed** $(\delta, \Omega) = (3 \times 10^{-6}, 3)$**, late steps** $L = 24$**.** Coverage reported as fractions. QKV-mean averages per-step similarities across $\{Q, K, V\}$; QKV-max takes a tokenwise maximum before rebuilding $P^{(t)}$. Row-wise best is in **bold**.

| Dataset | Q | K | V | QKV-mean | QKV-max |
|---|---|---|---|---|---|
| GSM8K | **0.984** | 0.234 | 0.219 | 0.969 | **0.984** |
| MATH500 | 0.984 | **1.000** | 0.000 | 0.984 | 0.813 |
| Sudoku | **0.375** | 0.094 | 0.125 | **0.375** | 0.344 |
| Countdown | 0.312 | 0.016 | **0.953** | 0.250 | 0.922 |

## H  ABLATION SCRIPTS AND REPRODUCIBLE WORKFLOW

**Scope.** This appendix documents the small set of scripts used to reproduce the projection ablations and aggregates. The pipeline is training-free: RRMs are derived from saved training artifacts (optimizer or adapter weights), and inference traces are collected once per projection.

**run_readout_ablation.py.** Python orchestrator with three subcommands: *collect* (run the extended collector to save per-step traces), *analyze* (compute AC, LSCR, and certified coverage at fixed $(\delta, \Omega)$ and $L$), and *summarize* (merge analyzer outputs into CSVs for tables).

**trace_ablation_orchestrator.py.** All-in-one driver that adds *prep* (standardize RRMs into `rrms/final_block_{Q,K,V}.json` from existing JSON or `.npy`) to *collect/analyze/-summarize* across datasets and projections.

**`build_rrms_from_checkpoints.py`.** Build RRM vectors from saved training artifacts. *Optimizer mode* uses AdamW `exp_avg` on LoRA-$B$ when a parameter-name map is present. *Weights mode* accumulates squared deltas of LoRA-$B$ across checkpoints (e.g., `adapter_model.safetensors`) and takes row-$\ell_2$.

**`build_single_rrm_from_adapter.py`.** Single-snapshot extractor: computes row-$\ell_2$ of LoRA-$B$ from one adapter file—used when only a single adapter snapshot is available.

**`merge_readouts_from_traces.py`.** QKV aggregators: given per-projection traces with per-step similarities/probabilities, aggregate (mean or tokenwise max) across $\{Q, K, V\}$ and rebuild $P^{(t)}$; writes analyzer-compatible merged traces (QKV-mean/max).

**Workflow (fixed** $(\delta, \Omega) = (3 \times 10^{-6}, 3)$**, $L$=24).** *(1) Prepare RRMs:* `prep` from an existing JSON or build from adapter weights (final block 31 for $K/V$). *(2) Collect traces:* datasets $\times$ $\{Q, K, V\}$ with shared prompts. *(3) (Optional) Aggregates:* run `merge_readouts_from_traces.py` to form QKV-mean/max. *(4) Analyze:* run `analyze` at $(3 \times 10^{-6}, 3)$ and late steps $L$=24. *(5) Summarize:* produce a single CSV for App. Table 4.

**Note on $O$/QKVO.** Our LoRA configuration attaches adapters to $Q/K/V$ only at the final block; no LoRA parameters were trained for the output projection, so $O$/QKVO variants are omitted.