# OpenReview forum: "Training Dynamics Explain Safe Early Exits in Diffusion Language Models"
_ICLR.cc/2026/Conference — Submitted to ICLR 2026_

### Official Review · Reviewer_x4Zd · 2025-10-27

**Soundness:** 1
**Presentation:** 1
**Contribution:** 1
**Rating:** 0
**Confidence:** 4

**Summary:**

The paper suggests an early stopping criteria when generating with a diffusion model. The paper is largely illegible. I could not understand the context, the proofs, or many of the statements.

**Strengths:**

This question of using some information from the optimization buffers to guide early stopping is interesting.

**Weaknesses:**

The context is lacking. I could not even say what was the underlying model, if this is some sort of discrete diffusion over tokens, or a more standard diffusion conditioned on tokens such as latent diffusion models?

Things become confusing when the details are defined in Section 2. Just to start, your notation for the Lora matrix is, at least, unusual. According to your shapes you must be using
$$W' = W + BA$$

The premise of this work, is that you can use some information from the optimization to help early stopping. For this, it seems you use $\bar{U}_B$ in eq (3). But this is simply the final low rank Lora update minus it's initialization, that is

$$B^0  - B^t  = \bar{U}_B$$

This is because the updates are additive, and you add together all the updates in eq (3). As such, you just need to remember how you initialized $B$ to recover $ \bar{U}_B$. There is no need to track and record $ \bar{U}_B$ during training.


In the next paragraph, what exactly is this $f_s^{(t)}$ activation on line 134? You need to define it. I'm guessing here it's

$$ f_s^{(t)} = \sigma(W + B^tA^t e_s)$$

Where $e_s$ is the $s$-th coordinate vector, and $\sigma$ is some activation function.

Then in Section 3, you state "margins $m_t$ are computed on the matched support $\mathcal{I}_t.$." You haven't defined $m_t$ or $\mathcal{I}_t$ yet.

I could not understand or appreciate any of your theoretical results in Lemma 3.1, Theorem 3.2 or Theorem 3.3. I can follow the proofs of repeatedly applying Pinsker's inequality, but what do these results mean? Of particular importance is this assumption in (9), why is this at all a fair assumption to make? How is this Markov operator $K_r$ in Assumption 3.1 defined? And what is its role here?

At this point, I got tired of trying to understand. This paper needs to be completely re-written. You need to clarify the context, and generally improve the writing.


Indeed, the paper is also littered with verbose confusing sentences such as

"Aggregating these patterns yields ... that encodes the model's computational structure".

What is a "model's  computational structure"? I don't think this means anything.


Another example

"Parameters with large value in $\bar{U}_B$ encode core reasoning patterns, while those with small or oscillating values contribute less to the learned capability."

This is such a confusing sentence, when what ypi meant was "By definition (2), large entries of $\bar{U}\_B$ correspond to large entires of the $B$."

And then again before eq(4) you state:

"We reduce $\bar{U}_B$ to a feature-aligned vector through row-wise energy". All you meant was, "we divide by the norm". To make matters worse, equation 4 does not match you description of $u$ later on. It seems you forgot to also normalize $u$. You then say immediately after this on line 128 "for subspace variants $U$ we orthonormalize columns". What are subspace variants? I really have no idea of what you are talking about here.

On line 144, what exactly do you mean by ""the set of visible tokens at step $t$"?

Good science writing needs to be very clear. Everything needs to be defined. Sentences should be short and simple. Even terse.

**Questions:**

See Weakness above.

---

### Official Review · Reviewer_1eW9 · 2025-11-01

**Soundness:** 3
**Presentation:** 1
**Contribution:** 3
**Rating:** 4
**Confidence:** 3

**Summary:**

The paper proposes using a Reasoning Representation Map (RRM) built from training traces of LoRA adapters to certify early exit during the denoising in diffusion language models.  The aim is to detect when a diffusion step’s prediction is certified stable so downstream steps can be skipped. The authors derive single-step and multi-step certificates and provide a practical stopping procedure called TRACE; experiments are on LoRA-finetuned adapters attached to final-block Q/K/V. The idea of leveraging training dynamics for inferece-time early-exit guarantees is interesting, but the scope, presentation, and gap between theory and observed dynamics require clarification.

**Strengths:**

1. **Novel, impactful idea:** Using training-time optimizer traces to produce inference time early-exit certificates for diffusion steps is original and has clear practical relevance for inference efficiency.

2. **Solid theoretical framing:** The paper provides rigorous statements that tie RRM-derived metrics to argmax preservation under stated assumptions.

3. **Concrete, implemented algorithm:** the introduced algorithm (TRACE) employs the certificates into a principled stopping rule.

4. **Good empirical exploration within scope:** Multiple readouts, calibration strategies, and ablations are explored, demonstrating thoughtful experimental design for the LoRA

**Weaknesses:**

1. **Mismatch between theory and observed dynamics.** In my understanding, the global stability result requires a contraction assumption (α < 1), but the empirical results about late-step estimates `α̂` are > 1 (amplifying), undermining the applicability of the theoretical guarantee.

2. **Limited evaluation scope  LoRA-only.** All experiments use LoRA adapter traces (final-block Q/K/V); it is unclear whether RRM signals and certificates generalize to full-parameter fine-tuning.

3. **Practical utility of multi-step (`Ω>1`) certification is limited.** Under the observed margins, multi-step certified coverage is negligible except for `Ω=1`, calling into question the usefulness of the multi-step theory in practice.

4. **Lack of a self-contained background and notation.** For better clarity and exposition it would be useful to have a section giving the background of discrete diffusion, and introduce the problem of early stopping at inference for such models.  Diffusion-specific terms (`denoising step t`, `unmasking`, `S_t`, `f_s^{(t`) are used without concise formal definitions, which reduces clarity of the exposition.

5. **TRACE algorithm relegated to appendix.** The main practical procedure (Alg.1) appears only in the appendix; a compact pseudocode or algorithm summary in the main text is completely lacking and hinders the exposition.

6. **Global RRM averaging may blur late-phase signals.** The RRM is a full-training average; averaging early and late updates might wash out late-emerging capability signals. No time-windowed RRMs are evaluated.

7. **Runtime costs underspecified.** Storage is reported as small, but per-token inference overhead (similarity/KL computations, hooks) and production deployment costs are not quantified

**Questions:**

1. The global theorem assumes α < 1 but reported `α̂` values exceed 1. Can the authors elaborate on this mismatch?

2. How dependent is the RRM signal on the LoRA low-rank structure and attachment point (final-block Q/K/V)? What are the main obstacles to applying the method to full-parameter fine-tuning?

3. Given that certified coverage for `Ω>1` is negligible under the current setup, do the authors see concrete analytic or practical avenues to tighten bounds so multi-step certificates become useful in practice?

4. It would be very useful if the authors could add a short, self-contained background section that formally defines masked diffusion models and the problem of early stopping together with the notions of `t` (denoising step), “unmasking,” `S_t`, `I_t = S_{t- ∩ S_t}`, and activations `f_s^{(t)}`.

5. The authors could consider moving a compact version of Algorithm 1 (TRACE) into the main text (or present a one-paragraph summary) so readers can immediately link theorems to the stopping rule.

6. Have the authors evaluated RRMs computed from time-restricted windows (e.g., last 10% of training) or exponentially weighted averages that emphasize late training?

7. Could the authors provide a short overhead analysis: per-token latency of TRACE similarity/KL computations, recommended snapshot frequency during training for effective RRMs, and costs for deploying this at LLM scale.

---

### Official Review · Reviewer_X8LB · 2025-11-01

**Soundness:** 2
**Presentation:** 2
**Contribution:** 3
**Rating:** 4
**Confidence:** 3

**Summary:**

This manuscript proposes a novel method, Training-Refined Adaptive Computation Exit (TRACE), to reduce the computational cost of diffusion language models more safely and efficiently. In detail, it formalizes Reasoning Representation Maps (RRM) from optimizer trajectories and defines Representational Alignment Distributions(RAD) for inference. Theoretically, they prove the safety of TRACE by establishing conditions under which predictions remain invariant. Empirically, they validate their efficiency on four reasoning benchmarks.

**Strengths:**

- The structure is well-organized, and the writing is easy to follow.

- This manuscript is interesting and novel, as it introduces RRM and RAD terminologies. Although there have been works studying the early exit strategy, this manuscript tries to figure out the safety guarantee and prove the guarantee.

**Weaknesses:**

- Experiments are limited. Only four experiments are conducted on reasoning tasks. Would it be possible to generalize to other tasks, or how it behave on other tasks?

- This manuscript only makes comparisons with the full-load baseline(no early exit). Are there any comparisons with other denoising methods or early-exit methods?

- The detailed TRACE algorithm is presented in the Appendix, though it's claimed to be light, I suggest an analysis on the computation overhead included would be better.

**Questions:**

I'm interested in the TRACE method, and I see that the authors mentioned in the related work their work extends insights of Frankle & Carbin(2019) and Fort & Ganguli(2019). Could you be more specific about this connection?

---

### Official Review · Reviewer_WbgQ · 2025-11-05

**Soundness:** 2
**Presentation:** 1
**Contribution:** 2
**Rating:** 2
**Confidence:** 3

**Summary:**

The paper at hand studies the link between learning dynamics and representations in the context of finetuning diffusion language models. The authors propose new metrics, and design an algorithm that provides early-exit guarantees on the number of diffusion steps.

**Strengths:**

The paper is thought-provoking in that it highlights a key fact: often, the training signal is not used to guide inference. The authors explore this link in the context of discrete diffusion, which is a nice idea. I appreciate that the paper combines theory and experiments, and that the mathematical derivations are formal and presented with precise notation.

**Weaknesses:**

This paper is challenging to read for several reasons.

1) The authors don't really introduce discrete diffusion, the discussion jumps directly into the maths and the definitions
2) There is very little intuition around the quantities and the results presented. Associating big names, such as "Reasoning Representation Map," can harm your presentation if there is no precise delivery of what this means.
3) The paper positions itself, right from the start, directly to a small set of people (finetuning discrete diffusion). This is fine, but the presentation is not accessible to a general ICLR reader right from line 1.
4) The last 4 pages are, to my eyes, just a "list of things"; the discussion is very fragmented. I do not understand why the authors decided to put this here and the main algorithm in the appendix.

The issues above point to more general problems:
1) It is not clear why the authors are solving these niche problems: what are your motivations? I can see a few but this is presented more as a theoretical exercise.
2) The paper is FULL of arbitrary definitions (with big names attached to those quantities), assumptions, and claims. An example of this is Section 3.3. -- the core of the paper: the authors provide a result under a strong assumption that is not discussed. The result is not commented.
3) Another example, perhaps the main one, is that of the definition of P in equation 6: why is this a critical quantity to track? Why does this become a probability distribution? Which weights (QKV) are you talking about? I simply dont understand this.
4) Related to the point above, I do not understand the need for talking about training dynamics and Adam. Equation 4 essentially represents how the weights changed in absolute value along training. The meaning of equation 5 to me is still mysterious. It is unclear to me why this is a good idea -- specifically because you are taking an inner product with a vector of all positive quantities... looks strange to me.

**Questions:**

See above. It is possible I did not understand some points. Unfortunately the presentation did not help.

---

### Meta-Review · Area_Chair_EXAr · 2025-12-16

**Summary:**

The paper studies the link between learning dynamics and representations in the context of finetuning diffusion language models, and proposes Training-Refined Adaptive Computation Exit (TRACE), to reduce the computational cost of diffusion language models more safely and efficiently. However, this paper has several issues, e.g., novelty, missing related work on discrete diffusion, weak motivation and intuition. All of the reviewers vote for rejection. I suggest to reject this submission.

**Reviewer Concerns:**

The authors didn't provide rebuttal.

**Reviewer Scores:**

The authors didn't provide rebuttal.

---

### Decision · Program_Chairs · 2026-01-26

Reject